# Goat *Pleomorphic Adenoma Gene 1* (*PLAG1*): mRNA Expression, CNV Detection and Associations with Growth Traits

**DOI:** 10.3390/ani13122023

**Published:** 2023-06-18

**Authors:** Qian Wang, Zhenyu Wei, Haijing Zhu, Chuanying Pan, Zhanerke Akhatayeva, Xiaoyue Song, Xianyong Lan

**Affiliations:** 1Key Laboratory of Animal Genetics, Breeding and Reproduction of Shaanxi Province, College of Animal Science and Technology, Northwest A&F University, Yangling 712100, China; 2Shaanxi Provincial Engineering and Technology Research Center of Cashmere Goats, Yulin University, Yulin 719000, China; 3Life Science Research Center, Yulin University, Yulin 719000, China

**Keywords:** *pleomorphic adenoma gene 1* (*PLAG1*) gene, mRNA expression, copy number variation (CNV), growth traits, goats

## Abstract

**Simple Summary:**

The *pleomorphic adenoma gene 1* (*PLAG1*) gene has attracted a lot of attention as a gene related to myogenesis. The *PLAG1* gene is a key gene for growth traits in sheep, cattle, pigs, and chickens. However, it has also become the focus of research to determine whether it is the dominant gene affecting growth traits in goats. To address this problem, a bioinformatics analysis, mRNA expression (n = 6), and identification of copy number variants (CNVs) (n = 224) of *PLAG1* were conducted in Shaanbei white cashmere (SBWC) goats. The findings indicated that the gene had a large number of conserved motifs, the expression level of the gene was higher in fetal goats than in adult goats, and all CNV loci were found to be associated with goat growth traits (*p* < 0.05), making them effective molecular marker loci for goat breeding.

**Abstract:**

The *pleomorphic adenoma gene 1* (*PLAG1*) gene, as the major gene responsible for growth, plays a vital role in myogenesis. Meanwhile, the relationship between copy number variation (CNV) of this gene and growth traits in goats remains unclear. Therefore, this study investigated four aspects: bioinformatics analysis, mRNA expression (n = 6), CNV detection (n = 224), and association analysis. The findings indicated that the gene had a large number of conserved motifs, and the gene expression level was higher in fetal goats than in adult goats. Three CNV loci were selected from the database, among which CNV1 was located in the bidirectional promoter region and was associated with goat growth traits. CNV analysis showed that CNV2 and CNV3 of the *PLAG1* gene were associated with growth traits such as body weight, heart girth, height at hip cross, and hip width (*p* < 0.05), with CNV1 loss genotype being the superior genotype, and CNV2 and CNV3 median and gain genotypes of being superior genotypes. This finding further confirms that the *PLAG1* gene is the dominant gene for growth traits, which will serve as theoretical guidance for goat breeding.

## 1. Introduction

As an important economic indicator, an animal’s growth performance is related to national living standards. In addition, animal body size is a key factor in differences in growth rate, energy metabolism, and body composition [1], which are regulated by multiple genes. Therefore, the search for the key genes controlling growth traits is the focus of researchers. In recent years, copy number variation (CNV), as a molecular marker for marker-assisted selection (MAS), can efficiently and easily select candidate genes related to the growth traits of animals, making a major contribution to animal breeding [2].

The *pleomorphic adenoma gene 1* (*PLAG1*) gene is a zinc-finger transcription factor, which is involved in cell proliferation by directly regulating multiple target genes and is the main locus determining the height and weight of animals [3]. Mice with the *PLAG1* gene knockout exhibited phenomena such as growth retardation and decreased fertility [4]. Studies have shown that the *PLAG1* gene upregulates phosphorylation of *Cyclin D1, p-Akt, Akt, (p)-PI3K, PI3K*, and *CDK2*, inhibits the expression of *p21* and *p27* to enhance myoblasts proliferation, and increases the expression of *Bcl-2* and *Bcl-xL* to suppress apoptosis [5]. Numerous experiments have demonstrated that the *PLAG1* gene is significantly associated with growth traits in livestock and poultry. Genome-wide association analysis reported that the *PLAG1* gene was associated with the body weight of broiler chickens [6], the shank weight of Simmental cattle [7], and the skeletal size of Japanese Black steers [8]. In addition, the insertion/deletion (InDel) in the *PLAG1* gene was significantly associated with the chest circumference and chest width of Luxi blackhead sheep [9] and Chinese cattle [10,11]. Furthermore, the single nucleotide polymorphism (SNP) in *PLAG1* was significantly associated with body size [12] and meat weight [13] of cattle, as well as with the number of vertebrae in pigs [14].

In addition, the *PLAG1* gene has been identified as a key gene involved in disease development, nervous system development, and reproductive function in animals. Previous studies have reported that the *PLAG1* gene was a key regulator of lung cancer [15], leiomyosarcoma [16,17], ovarian cancer [18], and rhabdomyosarcoma [19]. Hypermethylation of *PLAG1* has been associated with male infertility [20], and the *PLAG1* gene deficiency can cause abnormal curling of the epididymis [21], impair sperm movement [22], and reduce fertility [23]. Additionally, the *PLAG1* gene regulates neuronal gene expression and neuronal differentiation [24].

Although the *PLAG1* gene has been studied in myogenesis, CNV has not been reported in goats. Therefore, this study screened three CNV loci from the database, and Shaanbei white cashmere (SBWC) goats were taken as experimental subjects. The function of the *PLAG1* gene was explored from four aspects: gene sequence analysis, mRNA expression, CNV detection, and association analysis using bioinformatics analysis and quantitative real-time PCR (qRT-PCR). The results obtained can serve as a theoretical foundation for the MAS breeding of goats.

## 2. Materials and Methods

### 2.1. Animal Welfare Explanation

The collection of samples was conducted in accordance with China’s national standards: the Guidelines on Welfare and Ethical Review for Laboratory Animals (GB/T 35892-2018), and the experiments were approved by the Regulations on the Administration of Experimental Animals in Northwest A&F University (NWAFU-314020038).

### 2.2. Identification and Bioinformatics Analysis of the PLAG1 Gene

The *PLAG1* gene sequences (Human (NC_000008), macaque (NC_041761.1), mouse (NC_000070), rat (NC_051340), pig (NC_010446), cattle (NC_037341), goat (NC_030821), sheep (NC_056062), and chicken (NC_052533)), and protein sequences (Human (XP_016869065.1), macaque (XP_005563402.1), mouse (XP_036020164.1), rat (XP_017448710.1), pig (XP_020945013.1), cow (XP_005215489.1), goat (XP_017913986.1), sheep (XP_042110165.1), and chicken (NP_001385307.1)), were derived from the NCBI (https://www.ncbi.nlm.nih.gov/, accessed on 10 May 2022) database. The conserved protein motifs of the *PLAG1* gene were predicted using the online analysis tool MEME (http://meme-suite.org/index.html, accessed on 10 May 2022), where the parameter was set to maximum, the number of motifs was 15, and the length of motifs ranged from 6 to 200 aa. The gene structure and motifs were visualized and further tuned using the online platform Evolview (https://www.evolgenius.info/evolview/#login, accessed on 10 May 2022).

We used protein sequences of the human, macaque, mouse, rat, pig, cattle, goat, sheep, and chicken *PLAG1* gene to construct an evolutionary tree. The maffT software (http://mafft.cbrc.jp/alignment/server/large.html, accessed on 10 May 2022) was used for multiple sequence alignment of this gene (parameter: --localpair--maxiterate 1000) [25], and then the maximum likelihood (ML) method of IQ-TREE (http://www.iqtree.org, accessed on 10 May 2022) was used to construct the evolutionary tree, where the number of bootstraps was set to 1000. The Evoview online tool was used to analyze the evolutionary tree [26].

### 2.3. Extraction of Genomic DNA and RNA

Goat ear tissue (n = 224) from adult female goats (2–3 years old) was randomly selected for DNA extraction, and other tissues, including liver, spleen, lung, kidney, and muscle from female fetus (n = 9) and female adult (1.5 years) (n = 9) goats were collected for RNA extraction. All tissue samples were from the Yulin breeding base in Shaanxi province. Female goats were randomly selected to be raised in the same environment and under the same management conditions, and there is no kinship between individuals. Then, tissue samples were stored in 70% ethanol at −80 °C. Tissue RNA was extracted using the Trizol method and then reversely transcribed into cDNA using PrimeScript™ RT Reagent Kit (Takara Biotech, Dalian, China) [27]. A high salt extraction method was used to extract DNA from ear tissues [28]. The OD_260/280_ ratios of DNA and RNA samples were measured by a NanoDrop™1000 Spectrophotometer (Thermo Scientific, Waltham, MA, USA), and OD_260/280_ values were 1.6–1.8 for all DNA samples and 1.8–2.0 for RNA samples, finally diluted to 20 ng/uL and at −20 °C.

### 2.4. Designing of Primers

CNV loci were retrieved from the Animal Omics database (http://animal.nwsuaf.edu.cn/, accessed on 19 March 2022) [29], and we found three CNV loci of the *PLAG1* gene in goats. Based on reference sequences (NC_030821.1), the primers for mRNA expression and CNV detection were then designed using the Primer-blast tool in the NCBI database (https://www.ncbi.nlm.nih.gov/tools/primer-blast/, accessed on 19 March 2022) and the Primer Premier 5.0 software (Thermo Scientific, Waltham, USA) (Table 1). The *GAPDH* gene was used as the reference gene for mRNA expression, and the *MCIR* gene was utilized as the reference gene for CNV detection. GAPDH and MC1R were referred to in a previous article [30,31].

### 2.5. Detection of the PLAG1 Gene mRNA Expression Levels

The qRT-PCR method was used to detect the expression level of the *PLAG1* gene in fetal female goat tissues and adult female goat tissues. SYBR Premix Ex TaqTMII (Takara Biotech, Dalian, China) was used for detection, and the total detection system and amplification steps were based on a previous study [32]. The total system was 10 μL, including 5 μL of SYBR Premix Ex TaqTMII, 0.5 μL of F primer, 0.5 μL of R primer, and 4 μL of cDNA. The amplification steps were 45 s at 95 °C, 40 cycles of 15 s at 95 °C, and 60 °C for 45 s. The 2^−δδCt^ method was used to analyze data [30].

### 2.6. CNV Genotyping of the PLAG1 Gene

After testing the primers through the DNA pool, a total of 224 SBWC goat samples were used to detect CNV loci. The total detection system was 10 μL, including 5 μL of SYBR Premix Ex TaqTMII, 0.5 μL of F primer, 0.5 μL of R primer, 1 μL of DNA, and 3 μL of ddH_2_O. The amplification system and steps of qRT-PCR were as described above, i.e., 45 s at 95 °C, 40 cycles of 15 s at 95 °C and 60 °C for 45 s. The result was calculated using the 2^−δδCt^ method [33]. If the 2^−δδCt^ = 1, it was the median genotype, the 2^−δδCt^ > 1 was the gain genotype, the 2^−δδCt^ < 1 was the loss genotype.

### 2.7. Statistical Analyses

The frequency of CNV genotypes was analyzed using the chi-square test (χ^2^). After that, the association between the genotypes of CNVs and growth traits was analyzed by the Analysis of Variance (ANOVA) test in SPSS 26.0 (IBM, USA). When *p* < 0.05, the presence of an association between CNV and body weight, body height, height at hip cross, body length, hip width, heart girth, and other growth traits was established, and the line model was referred to a study by Yang et al. [34]. Where Y_ijk_ = α_i_ + β_j_ + e_ijk_ + u acts as a linear model, Y_ijk_ is the evaluation of growth traits at the i level of fixed factor age (α_i_) and j level of fixed factor genotype (β_j_), u is the overall mean, and e_ijk_ is the random error.

## 3. Results

### 3.1. Bioinformatics Analysis of PLAG1 Gene

Sequence analysis of the *PLAG1* gene using an online platform revealed that the *PLAG1* gene contains five conserved protein motifs (Figure 1a). In addition to this, multiple sequence alignment and evolutionary analysis of nine representative species, including pig, cattle, goat, sheep, and chicken, showed that goat had the highest sequence homology with sheep and cattle, and the pig was the closest relative to ruminants among several domestic animals, while the chicken was the most distant relative of these species (Figure 1b).

### 3.2. mRNA Expression Levels of Fetal and Adult Goats

To preliminary explore the *PLAG1* gene expression during different periods of goat development, qRT-PCR was used to find that the *PLAG1* gene in goat tissue was also expressed in fetal and adult goats, and the *PLAG1* gene expression was high in lung, kidney, spleen, liver, and muscle in fetal goats. Moreover, with growth and development, the relative expression of adult goats was relatively stable compared with fetal goats, while the expression levels decreased in the spleen, kidney, lung and muscle (Figure 2, Appendix A). It has been suggested that the *PLAG1* gene may be involved in the development of multiple tissues in animals.

### 3.3. Frequency of CNV Genotypes

CNVs of the *PLAG1* gene were found in the database, and CNV1 covers both the first exon and the bidirectional promoter of the *PLAG1* gene and the upstream *CHCHD7* gene, which was detected by comparing the gene sequence annotation information with mutation location information in the database (Figure 3).

The large-scale detection revealed that the percentage of CNV1 loss and gain genotypes was 42.47%, respectively, and the median genotype was 15.07%. In CNV2 and CNV3, the number of individuals with the median genotype (CNV2 = 153, 70.18%; CNV3 = 155, 74.52%) was greater than those with the loss genotype (CNV2 = 15, 6.88%; CNV3 = 10, 4.81%) and the gain genotype (CNV2 = 50, 22.94%; CNV3 = 43, 20.67%) (Figure 4, Appendix A), and the genotypes of the three CNV loci were not significantly different (*p* = 0.252).

### 3.4. Association Analyses between CNV and the Goat PLAG1 Gene

Association analysis showed that three CNV loci were associated with growth traits in SBWC goats. Among them, CNV1 was significantly associated with body weight (*p* = 2.701 × 10^−8^), body height (*p* = 0.002), height at hip cross (*p* = 8.000 × 10^−6^), body length (*p* = 0.008), hip width (*p* = 8.252 × 10^−8^), heart girth (*p* = 0.001), and cannon circumference (*p* = 4.660 × 10^−4^) (Table 2). CNV2 was significantly associated with body weight (*p* = 4.023 × 10^−3^), height at hip cross (*p* = 0.002), hip width (*p* = 2.000 × 10^−6^), heart girth (*p* = 3.270 × 10^−4^), and chest depth (*p* = 0.013) (Table 3). CNV3 was significantly associated with body weight (*p* = 0.020), height at hip cross (*p* = 0.003), body length (*p* = 0.022), hip width (*p* = 1.540 × 10^−4^), heart girth (*p* = 3.480 × 10^−4^), and cannon circumference (*p* = 0.020) (Table 4).

Additionally, we found that in CNV1, individuals with the loss genotype were superior to those with median and gain genotypes in terms of growth traits (Table 2), but the individuals with CNV2 and CNV3 with the median and gain genotypes were better than those with the loss genotypes (Table 3 and Table 4).

## 4. Discussion

The *PLAG1* gene is a key gene affecting animal growth [3]. Relevant studies have shown that the *PLAG1* gene plays an important role in growth regulation in mice [4], broiler chickens [6], Simmental cattle [7], Japanese Black steers [8], Luxi blackhead sheep [9], Chinese cattle [10,11], and other domestic animals by enhancing myoblasts proliferation and suppressing apoptosis [5]. Amino acid sequence alignment revealed that the sequences of this gene in cattle, sheep, and goats have high similarity [31]. In this study, the phylogenetic tree of the *PLAG1* gene showed a high degree of homology among cattle, goats, and sheep. Thus, given the importance of the *PLAG1* gene in growth performance in cattle, we hypothesized that it also plays a crucial role in goat growth traits.

Moreover, understanding the expression of the gene in various tissues is very important to explore its function. The results showed that the expression level of this gene was reduced in adult goats compared to fetal goats, which suggests that this gene can potentially be expressed in embryos and was consistent with previous reports [35,36,37]. The reason for this phenomenon is that this gene is necessary for the activation of other transcription factors and embryo development [38].

To further explore the function of this gene, we identified three CNVs from the database, in which CNV1 spans the *PLAG1* gene and the *CHCHD7* gene and is located in the bidirectional promoter region, which has a significant impact on the body size of livestock [39,40]. Among the three CNVs, CNV1 had the lowest proportion of median genotypes: the gain genotype and loss genotype accounted for an equal proportion. While CNV2 and CNV3 had the highest proportion of median genotypes, the loss genotype accounted for the smallest proportion. This may be due to the fact the mutation sites are located in different locations on the chromosome and have different effects on transcription factors, so they show different genotypes [41]. The results of the association analysis showed that CNVs were associated with the growth traits of goats (*p* < 0.05). CNV1 was significantly associated with body weight, body height, height at hip cross, body length, hip width, heart girth, and cannon circumference. CNV2 was significantly associated with body weight, height at hip cross, hip width, heart girth, and chest depth. CNV3 was significantly associated with body weight, height at hip cross, body length, hip width, heart girth and cannon circumference. In addition, the loss genotype of CNV1 was the dominant genotype, while the median and gain genotypes of CNV2 and CNV3 were superior genotypes. This may be due to artificial selection and genetic drift. Previous studies have also reported that SNP, InDel and CNV within *PLAG1* can affect growth traits in cattle and sheep [9,10,11,12], which are consistent with the results of this study.

Although the growth trait is a multi-gene controlled trait, the influence of CNV on the genetic trait of livestock and poultry has also been proven. Li et al. found that CNVs of 52 candidate genes were significantly associated with the tail type, tail length, and tail fat of Hulunbuir sheep [42]. In Hu sheep and Small Tail Han sheep, the CNVs of the *SHE* gene were associated with chest circumference and chest width [43], and the CNVs of the *BAG4* gene were associated with body slanting length and body height [44]. In addition, Liu et al. revealed the population differentiation of CNVs in different geographical regions, which may reflect the population history of different goat breeds [45]. In the *MYLK4* gene of Guizhou white goats, CNV had significant differences in weight, height, and body length [46]. Previous studies have shown that CNVs of *EIF4A2* [47], *GAL3ST1* [48], and *DYNC1I2* [49] genes were associated with growth traits in Chinese cattle. Skinner et al. supported the hypothesis that the genomes of evolutionarily distant species share CNV regions [50]. In conclusion, CNVs have an impact on animal traits, and CNVs may reflect the evolutionary trend of a population.

From bacteria and archaea to plants and animals, CNVs with DNA sequence amplification and deletion exist in the genome of organisms [51]. Many studies have shown that this technique plays an important role in screening candidate genes and other aspects. In this study, CNV of the *PLAG1* gene will enrich the function of this gene and provides theoretical guidance for goat breeding. In the future, CNV will be a simple, efficient, and widely applicable molecular marker.

## 5. Conclusions

In this study, the *PLAG1* gene had a large number of conserved motifs, and mRNA expression of the *PLAG1* gene was higher in fetal tissue than in adult goats. The three CNVs were significantly associated with the growth traits of goats (*p* < 0.05). CNV1 was significantly associated with body weight, body height, height at hip cross, body length, hip width, heart girth, and cannon circumference. CNV2 was significantly associated with body weight, height at hip cross, hip width, heart girth, and chest depth. CNV3 was significantly associated with body weight, height at hip cross, body length, hip width, heart girth and cannon circumference. Meanwhile, the loss genotype of CNV1 was the dominant genotype, and the median genotype and gain genotype of CNV2 and CNV3 were the superior genotypes. These findings confirm that the *PLAG1* gene can be an essential functional candidate gene for growth traits, allowing more extensive use of MAS in goat breeding.

## Figures and Tables

**Figure 1 animals-13-02023-f001:**
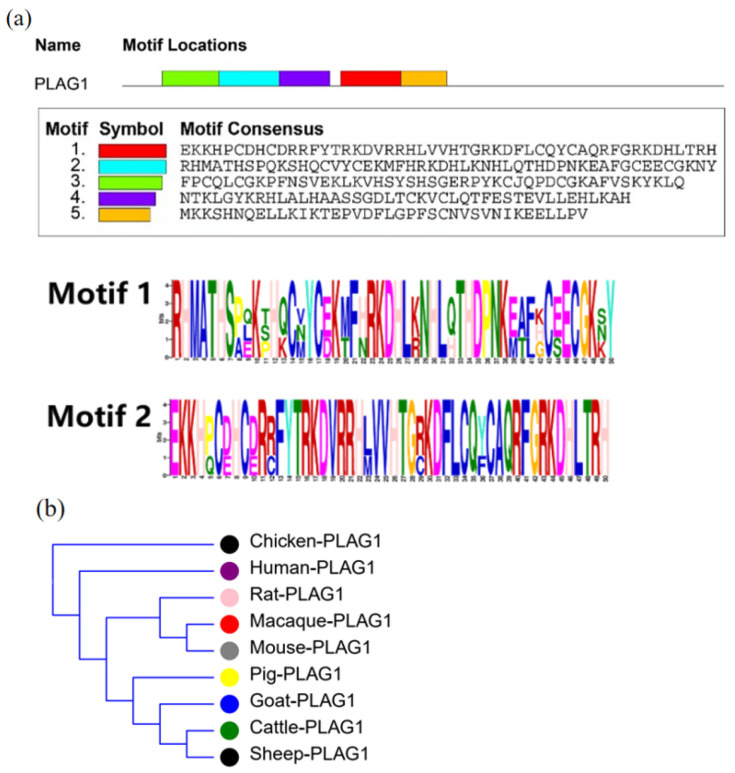
Identification and bioinformatics analysis of the *PLAG1* gene. **Note:** (**a**): Schematically displaying Motif1 and Motif2 of the PLAG1; (**b**): gene evolution tree analysis of *PLAG1* in different species.

**Figure 2 animals-13-02023-f002:**
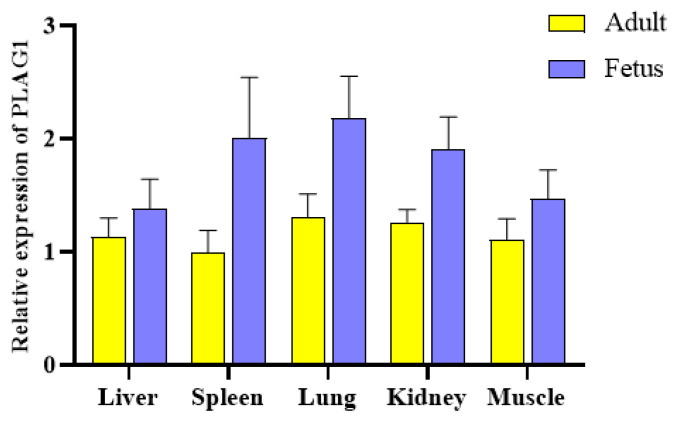
*PLAG1* mRNA expression levels in different tissues of fetal and adult goats.

**Figure 3 animals-13-02023-f003:**
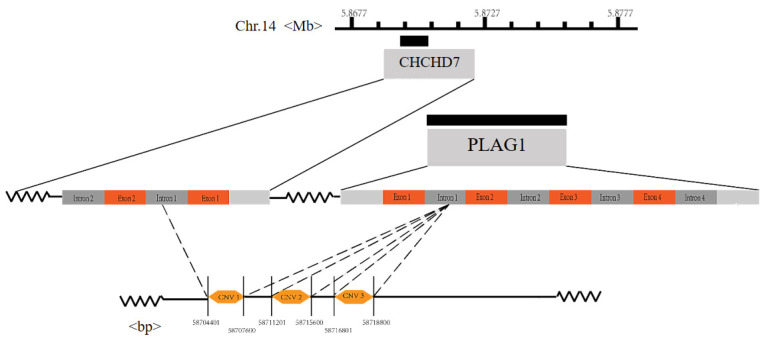
Schematic diagram of the *PLAG1* gene CNV loci on the genome. **Note:** Chr. 14 means Chromosome 14.

**Figure 4 animals-13-02023-f004:**
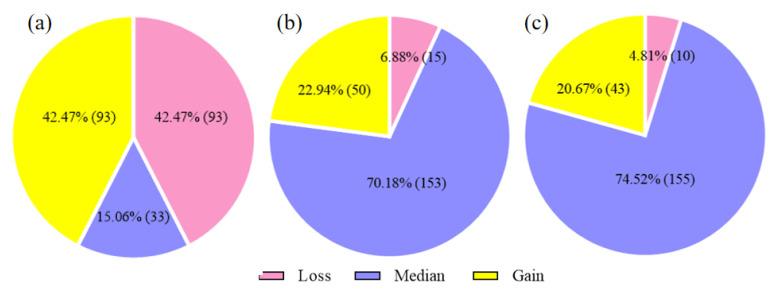
Frequency distribution of different genotypes of three CNV polymorphisms of the *PLAG1* gene in SBWC goats. **Note:** (**a**): CNV1; (**b**) CNV2; (**c**) CNV3. The total number of CNV1 was 219, CNV2 was 218, and CNV3 was 208.

**Table 1 animals-13-02023-t001:** The primer information of mRNA expression and CNVs.

Primer Names	Primer Pairs (5′-3′)	Tm (°C)	Length (bp)
PLAG1-mRNA	F: ATTCCGCGGGCGGTGTAAA	62.61	298
	R: TCACCAGGAATGACAGTGGC	59.96	
GAPDH	F: AAAGTGGACATCGTTGCCAT	60.04	116
	R: CCGTTCTCTGCCTTGACTGT	59.97	
CNV1	F: CTTTCGACGGCACCCAAGAA	60.88	199
	R: TTGCCGTTGCGCTTCCT	60.26	
CNV2	F: CTCAAACATGAATTTGCTCCCC	58.15	250
	R: AGACTGCTTCCACCACTCATA	58.46	
CNV3	F: CGTATAGGGGAAGCTCAG	50.20	192
	R: GTTTGGATGTCTTCTATTTCTT	50.10	
MC1R	F: GGCCTGAGAGGGGAATCACA	61.27	126
	R: AGTGGGTCTCTGGATGGAGG	60.33	

**Table 2 animals-13-02023-t002:** Association analyses between growth traits and the CNV1 in SBWC goats.

Trait-Types	Typic Frequencies (AVG ± SE)			*p*-Values
	Loss	Median	Gain	
**body weight (kg)**	**52.42 ± 1.58 ^A^ (n = 70)**	**43.35 ± 3.38 ^AB^ (n = 17)**	**40.68 ± 1.05 ^B^ (n = 65)**	**2.701 × 10^−8^**
**body height (cm)**	**57.51 ± 0.50 ^A^ (n = 93)**	**56.89 ± 0.81 ^AB^ (n = 33)**	**55.15 ± 0.42 ^B^ (n = 92)**	**0.002**
**height at hip cross (cm)**	**61.17 ± 0.54 ^A^ (n = 92)**	**59.83 ± 0.95 ^AB^ (n = 33)**	**57.95 ± 0.38 ^B^ (n = 91)**	**8.000 × 10^−6^**
**body length (cm)**	**67.75 ± 0.51 ^A^ (n = 93)**	**64.77 ± 0.99 ^B^ (n = 33)**	**67.73 ± 0.51 ^A^ (n = 91)**	**0.008**
**hip width (cm)**	**17.37 ± 0.33 ^A^ (n = 87)**	**16.98 ± 0.61 ^A^ (n = 26)**	**15.11 ± 0.19 ^B^ (n = 77)**	**8.252 × 10^−8^**
**heart girth (cm)**	**90.00 ± 1.09 ^A^ (n= 93)**	**86.10 ± 1.60 ^B^ (n = 31)**	**84.70 ± 0.88 ^B^ (n = 92)**	**0.001**
**cannon circumference (cm)**	**8.45 ± 0.08 ^A^ (n = 93)**	**8.34 ± 0.11 ^A^ (n = 32)**	**8.03 ± 0.07 ^B^ (n = 93)**	**4.660 × 10^−4^**
chest width (cm)	20.66 ± 0.34 (n = 93)	20.00 ± 0.50 (n = 33)	19.71 ± 0.41 (n = 92)	0.183
chest depth (cm)	28.78 ± 0.28 (n = 93)	27.98 ± 0.53 (n = 32)	28.66 ± 0.30 (n = 92)	0.381

**Note:** Values with different letters (A, B) within the same row differ significantly at *p* < 0.01. AVG means average, and SE means standard error.

**Table 3 animals-13-02023-t003:** Association analyses between growth traits and the CNV2 in SBWC goats.

Trait-Types	Typic Frequencies (AVG ± SE)			*p*-Values
	Loss	Median	Gain	
**body weight (kg)**	**35.91 ± 3.68 ^b^ (n = 8)**	**48.50 ± 1.29 ^A^ (n = 105)**	**41.61 ± 1.62 ^B^ (n = 38)**	**4.023 × 10^−3^**
body height (cm)	56.07 ± 1.08 (n = 15)	56.86 ± 0.39 (n = 153)	55.65 ± 0.54 (n = 50)	0.254
**height at hip cross (cm)**	**56.93 ± 0.99 ^B^ (n = 15)**	**60.382 ± 0.42 ^A^ (n = 152)**	**58.12 ± 0.49 ^B^ (n = 49)**	**0.002**
body length (cm)	64.04 ± 1.71 (n = 14)	67.21 ± 0.38 (n = 153)	67.68 ± 0.75 (n = 50)	0.510
**hip width (cm)**	**13.94 ± 0.61 ^b^ (n = 9)**	**16.97 ± 2.98 ^A^ (n = 136)**	**15.16 ± 0.23 ^B^ (n = 44)**	**2.000 × 10^−6^**
**heart girth (cm)**	**82.40 ± 2.51 ^B^ (n = 15)**	**88.72 ± 0.79 ^A^ (n = 151)**	**83.18 ± 1.24 ^B^ (n = 50)**	**3.270 × 10^−4^**
cannon circumference (cm)	8.13 ± 0.14 (n = 15)	8.34 ± 0.06 (n = 153)	8.06 ± 0.10 (n = 50)	0.059
chest width (cm)	19.18 ± 0.90 (n = 15)	20.20 ± 0.26 (n = 153)	19.83 ± 0.53 (n = 50)	0.481
**chest depth (cm)**	**26.63 ± 0.57 ^c^ (n = 15)**	**28.61 ± 0.20 ^b^ (n = 152)**	**29.05 ± 0.51 ^a^ (n = 50)**	**0.013**

**Note:** Values with different letters (A, B, C/a, b, c) within the same row differ significantly at *p* < 0.01/*p* < 0.05. AVG means average, and SE means standard error.

**Table 4 animals-13-02023-t004:** Association analyses between growth traits and the CNV3 in SBWC goats.

Trait-Types	Typic Frequencies (AVG ± SE)			*p*-Values
	Loss	Median	Gain	
**body weight (kg)**	**35.37 ± 4.22 ^b^ (n = 6)**	**48.09 ± 1.31 ^a^ (n =104)**	**42.34 ± 1.60 ^b^ (n = 38)**	**0.020**
body height (cm)	55.60 ± 1.41 (n = 10)	56.67 ± 0.40 (n = 154)	55.99 ± 0.56 (n = 43)	0.685
**height at hip cross (cm)**	**57.55 ± 0.65 ^B^ (n = 10)**	**60.23 ± 0.43 ^A^ (n = 152)**	**58.04 ± 0.47 ^B^ (n = 43)**	**0.003**
**body length (cm)**	**63.65 ± 1.45 ^B^ (n = 10)**	**67.26 ± 0.39 ^ab^ (n = 153)**	**68.45 ± 0.81 ^a^ (n = 43)**	**0.022**
**hip width (cm)**	**14.50 ± 0.73 ^B^ (n = 6)**	**16.82 ± 0.26 ^A^ (n = 138)**	**15.25 ± 0.27 ^B^ (n = 42)**	**1.540 × 10^−4^**
**heart girth (cm)**	**82.70 ± 2.70 ^AB^ (n = 10)**	**88.36 ± 0.74 ^A^ (n = 154)**	**82.26 ± 1.47 ^B^ (n = 43)**	**3.480 × 10^−4^**
**cannon circumference (cm)**	**7.75 ± 0.25 ^b^ (n = 10)**	**8.35 ± 0.06 ^A^ (n = 155)**	**8.13 ± 0.10 ^Ab^ (n = 43)**	**0.020**
chest width (cm)	17.67 ± 1.03 (n = 10)	20.24 ± 0.24 (n = 154)	20.29 ± 0.69 (n = 43)	0.098
chest depth (cm)	27.25 ± 0.67 (n = 10)	28.44 ± 0.19 (n = 153)	29.48 ± 0.61 (n = 43)	0.062

**Note:** Values with different letters (A, B/a, b) within the same row differ significantly at *p* < 0.01/*p* < 0.05. AVG means average, and SE means standard error.

## Data Availability

There is no new data were created.

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
