# Peer review of "Goat Pleomorphic Adenoma Gene 1 (PLAG1): mRNA Expression, CNV Detection and Associations with Growth Traits"

_animals, 2023, doi:10.3390/ani13122023_

Round 1
Reviewer 1 Report (Previous Reviewer 2)
Authors have make improvements to the manuscript following all the comments made by the reviewers. It can be considered for further publication in the journal.
Author Response
Thank you for your review and best wishes to you.
Qian Wang (wangqian21028@163.com) (first author),
Ph.D. Xianyong Lan (lanxianyong79@126.com) (corresponding author)
College of Animal Science and Technology, Northwest A&F University, Yangling, Shaanxi, 712100, PR China.
Ph.D. Xiaoyue Song (songxiaoyue@yulinu.edu.cn) (corresponding author)
Shaanxi Provincial Engineering and Technology Research Center of Cashmere Goats, Yulin University, Yulin, Shaanxi, 719000, China;
Life Science Research Center, Yulin University, Yulin, Shaanxi, 719000, China.
Reviewer 2 Report (New Reviewer)
The manuscript aims to analyse PLKAG1 gene in Shaanbei white cashmere (SBWC) goats.
In general, the presented manuscript seems not to be the final version. Figures are duplicated and inclusions/exclusions of text are highlighted. Methods section must be improved to include important information and to avoid some questions.
Discussion is poor and the gene expression information is almost absent.
Considering this, I consider that the manuscript must be improved to better describe the findings.
Bellow I pinpoint some specific questions throughout the text that can help to increase comprehension.
2. Materials and methods
2.2. Identification and bioinformatics analysis of the PLAG1 gene
L87-88. "The PLAG1 gene sequences and protein sequences derived from the NCBI..."
Authors should present Accession numbers for the sequences used.
L94-95. "We used protein sequences of the PLAG1 gene in human, macaque, mouse, rat, pig..."
Authors should present Accession numbers for the sequences used.
2.3. Extraction of genomic DNA and RNA
L102-L104 "Goat ear tissue (n = 224) from adult female goats (2-3 years) were randomly selected"
Author should include sex information for the individuals sampled for RNA extraction and more information about fetal samples
L110-114 "The OD260/280 ratios of DNA and RNA samples were measured by a NanoDrop™1000 ..."
DNA samples were diluted to 20 ng/mL or 20ng/uL?
2.7. Statistical analyses
L141 Linear model? Authors should describe the model used in these analyses
3.2. mRNA expression levels of fetal and adult goats
L164 There are two completely different figure 2. I couldn't understand which one is the correct.
3.3. Frequency of CNV genotypes
L179 There are two figure 3. I couldn't understand which one is the correct. Additionally, I was confused with CNV structure. Author should clarify this information.
3.4. Association analyses between CNV and the goat PLAG1 gene
L188 Correlation or association analysis?
4. Discussion
L234-235 Whats the definition of dominant genotype?
Author Response
Dear reviewer:
Thank you for your suggestions on the manuscript. According to your suggestions, I have made corrections in the manuscript. The specific answers are as follows:
Point 1: L87-88. "The PLAG1 gene sequences and protein sequences derived from the NCBI..."
Authors should present Accession numbers for the sequences used.
Response 1: Thank you very much for your advice, we have supplemented the reference DNA sequence and protein sequence in the manuscript. Please check the revised manuscript in the attachment.
Point 2: L94-95. "We used protein sequences of the PLAG1 gene in human, macaque, mouse, rat, pig..."
Authors should present Accession numbers for the sequences used.
Response 2: Thank you very much for your advice, we have supplemented the reference protein sequence in the manuscript. Please check the revised manuscript in the attachment.
Point 3: L102-L104 "Goat ear tissue (n = 224) from adult female goats (2-3 years) were randomly selected"
Author should include sex information for the individuals sampled for RNA extraction and more information about fetal samples
Response 3: Thanks for your suggestion. The RNA was extracted from a female goat sample that had been modified in the manuscript. Please check the revised manuscript in the attachment.
Point 4: L110-114 "The OD260/280 ratios of DNA and RNA samples were measured by a NanoDrop™1000 ..."
DNA samples were diluted to 20 ng/mL or 20ng/uL?
Response 4: Thank you very much for your advice, DNA samples were diluted to 20ng/uL, We have revised it in the manuscript.
Point 5: L141 Linear model? Authors should describe the model used in these analyses
Response 5: Thank you for your proposal. We've described it in the manuscript, the “Where Yijk = αi + βj + ℇijk + μ acts as a linear model, Yijk is the evaluation of growth traits at the i level of fixed factor age (αi) and j level of fixed factor genotype (βj), μ is the overall mean and ℇijk is the random error.”
Point 6: L164 There are two completely different figure 2. I couldn't understand which one is the correct.
Response 6: This is probably because the document revision mode is turned on, and one of the images is previously modified.
Point 7: L179 There are two figure 3. I couldn't understand which one is the correct. Additionally, I was confused with CNV structure. Author should clarify this information.
Response 7: This is probably because the document revision mode is turned on, and one of the images is previously modified.
Point 8:L188 Correlation or association analysis?
Response 8: Thank you for your advice. We used association analysis. Changes have been made in the documentation.
Point 9:L234-235 Whats the definition of dominant genotype?
Response 9: Dominant genotype refers to the genotype carried by individuals who are more suitable for living conditions and production needs in actual production.
Kind thanks to you for your suggestions.
Sincerely Yours,
Qian Wang (wangqian21028@163.com) (first author),
Ph.D. Xianyong Lan (lanxianyong79@126.com) (corresponding author)
College of Animal Science and Technology, Northwest A&F University, Yangling, Shaanxi, 712100, PR China.
Ph.D. Xiaoyue Song (songxiaoyue@yulinu.edu.cn) (corresponding author)
Shaanxi Provincial Engineering and Technology Research Center of Cashmere Goats, Yulin University, Yulin, Shaanxi, 719000, China;
Life Science Research Center, Yulin University, Yulin, Shaanxi, 719000, China.
Round 2
Reviewer 2 Report (New Reviewer)
Authors presented a revised version answering all questions performed in the anterior version.
This manuscript is a resubmission of an earlier submission. The following is a list of the peer review reports and author responses from that submission.
Round 1
Reviewer 1 Report
The manuscript by Qian Wang, Zhenyu Wei, Haijing Zhu, Chuanying Pan, Zhanerke Akhatayeva, Xiaoyue Song and Xianyong Lan, entitled “Goat PLAG1: mRNA expression, CNV detection and associa-2 tions with growth traits”, describes the study of the PLAG1 gene (myogenesis related) in a local breed of Shaanbei white cashmere goats and its association to the growth traits in goats. To achieve the study's objectives, the authors used bioinformatics tools, analyzed PLAG1 mRNA expression, identified PLAG1 CNVs and performed association analysis.
The manuscript is understandable. However, there are some mistakes. English native speaker should read the text, because the text needs corrections. Some sentences need to be improved stylistically (e.g. sentence in lines 41-43).
There are also editorial errors, such as extra spaces (e.g. line 15, 18).
Simple Summary
Please correct the gene name abbreviation – should be written in italics.
Introduction is a good background for the study.
Methods
The description of the research material is inadequate. There is no information about the number of herds from which the material was taken, as well as about the possible relationship of the studied individuals. The kinship of individuals can significantly affect the results of the study.
Too few individuals were used for gene expression analysis.
The authors need to improve the description of the methodology regarding the reference genes used. The description is misleading. It is not clear which gene the authors used as a reference gene for gene expression and which one as a control for CNV. The reader can only realize this by reading the cited source publications.
Writing about the methodology the Authors wrote: “GAPDH and MC1R were utilized as reference genes. GAPDH and MC1R were referred to a previous article [31]”. I checked that citation. In the reference No 31 methodology the information about GAPDH and MC1R is limited to pointing to another, different publications that supposedly describes it, but I haven't checked further. If the authors cite another publication in the methodology, the citation should point to the publication in which the aspect of the methodology was originally and comprehensively described, not to a publication that refers to subsequent sources. This is an unprofessional act, and furthermore raises questions about the advisability of citation one's own previous work. How many more sources does the reader have to read to find a description of an experiment on gene expression and selection of the appropriate reference gene?
In conclusion, there is no proper description of the gene expression experiment.
Results
Figure 1: Motif 1 and Motif 2 are completely unreadable > please correct that.
Figure 3: The Authors must improve readability of the whole figure.
Discussion
Drawing any firm conclusions about why PLAG1 expression is lower in adult goats than in fetuses, based on a comparison of 3 samples in each group, is a misunderstanding. Gene expression depends on many different factors, and 3 specimens per study group is far too few to draw any confident conclusions. One can speculate about a trend based on such groups, but nothing more. Especially since the lack of a proper description of the methodology used raise quite a few doubts about the correctness of this part of the study.
The authors' association studies focus exclusively on linking CNVs to traits. Features used for association studies are mostly known polygenic features. Whether it is valid to draw conclusions about the influence of a single gene CNVs on such traits remains debatable. At the very least, the authors should mention in the discussion that traits from an association study are not simple traits depending on a single gene.
References
Among 46 reference, 8 were published before 2017 and the rest have been published in the last 5 years. However, at least 12 reference (about 26%; Reference No 9, 12, 27, 28, 29, 31, 32, 33, 35, 36, 37, 45) are self-citations. It is not necessarily engaging in unethical behavior, but closer scrutiny may be needed. It is widely accepted that self-citations should not exceed 20%, and in some journals even 15% or 10%. For example, why the Authors cite the manuscripts about deletion in PRND gene or PRNT gene expression? It raises doubts.
In general, the paper broaden the knowledge about the PLAG1 gene CNVs in goats. There is no proper description of the gene expression experiment. There are doubts to the references. The manuscript could possibly be published after the gene expression section has been completely improved, or without the gene expression section, after next peer review.
Author Response
Dear reviewer:
Thank you for your suggestions on the manuscript. According to your suggestions, I have made corrections in the manuscript. The specific answers are as follows:
Point 1: The manuscript is understandable. However, there are some mistakes. English native speaker should read the text, because the text needs corrections. Some sentences need to be improved stylistically (e.g. sentence in lines 41-43). There are also editorial errors, such as extra spaces (e.g. line 15, 18).
Response 1: Thank you very much for your advice, we have carefully checked the manuscript where redundant parts have been deleted and some errors have been corrected. Please check the revised manuscript in the attachment.
Point 2: Simple Summary: Please correct the gene name abbreviation – should be written in italics.
Response 2: Thank you very much for your advice, we've gone over the manuscript and corrected all the mistakes. Please check the revised manuscript in the attachment.
Point 3: Methods: The description of the research material is inadequate. There is no information about the number of herds from which the material was taken, as well as about the possible relationship of the studied individuals. The kinship of individuals can significantly affect the results of the study.
Response 3: Thanks for your suggestion, we have described sample information in the Materials and Methods section. “Goat ear tissue (n = 224) from adult female goats (2-3 years) and other tissues including liver, spleen, lung, kidney, muscle from fetus (n = 3) and adult (1.5 years) (n = 3) goats from Yulin breeding base in Shaanxi province, and there is no kinship between individuals.” Please see the attachment for details.
Point 4: Methods: too few individuals were used for gene expression analysis.
Response 4: Thank you for your proposal. Three individuals were selected from each fetal goat and adult goat to conform to the minimum biological replicate. In future studies, we will refine this problem and increase the biological replicates to more than 5.
Point 5: Methods: there is no proper description of the gene expression experiment.
Response 5: Thank you for your advice. We have given a detailed description of gene expression in the section of Materials and methods, and consulted and cited the relevant literature. “GAPDH was used as the reference gene for mRNA expression and MCIR was utilized as the reference gene for CNV detection. GAPDH and MC1R were referred to a previous article.” Please see the attachment for details.
Point 6: Results: Figure 1: Motif 1 and Motif 2 are completely unreadable > please correct that.
Figure 3: The Authors must improve readability of the whole figure.
Response 6: Thanks for your suggestions, Figures 1 and 3 have been recreated. Please see the attachment for details.
Point 7: Discussion: Drawing any firm conclusions about why PLAG1 expression is lower in adult goats than in fetuses, based on a comparison of 3 samples in each group, is a misunderstanding. Gene expression depends on many different factors, and 3 specimens per study group is far too few to draw any confident conclusions. One can speculate about a trend based on such groups, but nothing more. Especially since the lack of a proper description of the methodology used raise quite a few doubts about the correctness of this part of the study.
Response 7: Thank you for your suggestion. Three samples per group really cannot draw a positive conclusion. Therefore, we will redescribe the discussion part. Please see the attachment for modification information.
Point 8: Discussion: The authors' association studies focus exclusively on linking CNVs to traits. Features used for association studies are mostly known polygenic features. Whether it is valid to draw conclusions about the influence of a single gene CNVs on such traits remains debatable. At the very least, the authors should mention in the discussion that traits from an association study are not simple traits depending on a single gene.
Response 8: Thank you very much for your advice. We explained the problem in the discussion section as suggested. Please see the attachment for modification information.
Point 9: References: among 46 reference, 8 were published before 2017 and the rest have been published in the last 5 years. However, at least 12 reference (about 26%; Reference No 9, 12, 27, 28, 29, 31, 32, 33, 35, 36, 37, 45) are self-citations. It is not necessarily engaging in unethical behavior, but closer scrutiny may be needed. It is widely accepted that self-citations should not exceed 20%, and in some journals even 15% or 10%. For example, why the Authors cite the manuscripts about deletion in PRND gene or PRNT gene expression? It raises doubts.
Response 9: Thank you for your suggestion. We have carefully checked the references and reduced the self-citation rate to less than 20% according to the requirements of the journal. Please check the attachment for specific information.
Kind thanks to you for your suggestions.
Sincerely Yours,
Qian Wang (wangqian21028@163.com) (first author),
Ph.D. Xianyong Lan (lanxianyong79@126.com) (corresponding author)
College of Animal Science and Technology, Northwest A&F University, Yangling, Shaanxi, 712100, PR China
Ph.D. Xiaoyue Song (songxiaoyue@yulinu.edu.cn) (corresponding author)
Shaanxi Provincial Engineering and Technology Research Center of Cashmere Goats, Yulin University, Yulin, Shaanxi, 719000, China;
Life Science Research Center, Yulin University, Yulin, Shaanxi, 719000, China;

Reviewer 2 Report
Authors present an interesting bioinformatic approach of goat PLAG1 gene. They provide valuable information about the function of the certain genes. There are some missing information or gabs before any further publication, that to my opinion is crucial to be added by authors and which I think would be able for them to cover. Please see bellow my specific comments:
-Title: please avoid abbreviations.
-l. 27 what was the focus of bioinformatic analysis. please specify.
-l. 55 "significantly correlated". Please change to associated as you do not present statistical data.
-l. 86-89 specify why do you choose these parameters.
-l. 99 specify the age of adult goats and the treatment of tissues upon RNA extraction.
-l. 110-112. Relative quantification was based in both reference genes or in only one please specify. In addition specify the number of replicates used in Real Time PCR.
-l. 121 based on which reference gene?
-l 130 motifs--> protein motifs
-Figure 1 motifs with colored letters are only depicted for 1 and 2, not the rest.
-l 142 omit the second "was"
-l.155-158 add the p value in the comparisons you stated
-Please format correctly tables 2, 3 and 4 to present data according to the right categories.
-l 225 add p value after significantly
-l 206-208 how do you explain the equal proportion of the rest genotypes. Please specify.
-Authors could also add in the discussion section a comparative analysis of CNV with other examined species and discuss how evolutionary may assist the observed (if any) changes in the number. I think that this could help the strength of the manuscript.
Author Response
Dear reviewer:
Thank you for your suggestions on the manuscript. According to your suggestions, I have made corrections in the manuscript. The specific answers are as follows:
Point 1: Title: please avoid abbreviations.
Response 1: The corresponding error in the manuscript has been corrected. Please see the attachment for details.
Point 2: 27 what was the focus of bioinformatic analysis. please specify.
Response 2: Biological analysis can give us an in-depth understanding of the gene. For example, analysis software can visually predict motif and detect the presence of enriched motif, and then analyze whether there is a motif with specific binding of transcription factors, so as to determine whether the binding of this region will affect gene expression.
Point 3: 55 "significantly correlated". Please change to associated as you do not present statistical data.
Response 3: Thank you for your suggestion. We have revised it in the manuscript. Please see the attachment for the modification information.
Point 4: 86-89 specify why do you choose these parameters.
Response 4: We have no way of knowing in advance the actual number of domains in this sequence, so we set the parameter to the maximum, which is “the parameter was set to maximum, the number of motifs was 15, and the length of motifs range from 6 to 200 aa”. In addition, this is a common parameter used in maffT software for multi-sequence comparison. --localpair specifies the L-INS-I method with the highest accuracy for comparison, while the maximum number of 1000 iterations for the method based on accuracy priority is also a relatively strict comparison standard to ensure the accuracy of the results.
Point 5: 99 specify the age of adult goats and the treatment of tissues upon RNA extraction.
Response 5: We have described the sample information in detail in the manuscript. “Goat ear tissue (n = 224) from adult female goats (2-3 years) and other tissues including liver, spleen, lung, kidney, muscle from fetus (n = 3) and adult (1.5 years) (n = 3) goats from Yulin breeding base in Shaanxi province, and there is no kinship between individuals. Tissue RNA was extracted using the Trizol method, and then reversely transcribed into cDNA using PrimeScript™ RT Reagent Kit (TaKaRa Biotech Co. Ltd) [27]. Then, RNA samples were stored at −80 oC. High salt extraction method was used to extract DNA from ear tissues [28]. Methods for determining the quality of extracted RNA and DNA were referred to previous articles [29], finally diluted to 20 ng/mL and at -20 oC.” Please see the attachment for the modification information.
Point 6: 110-112. Relative quantification was based in both reference genes or in only one please specify. In addition specify the number of replicates used in Real Time PCR.
Response 6: Thank you for your suggestion. “GAPDH was used as the reference gene for mRNA expression and MCIR was utilized as the reference gene for CNV detection. GAPDH and MC1R were referred to a previous article. The total detection system and amplification steps were based on a previous study [33], 45 s at 95 oC , 40 cycles of 15 s at 95 oC and 60 oC for 45 s.” Please refer to the attachment for modification information.
Point 7: 121 based on which reference gene?
Response 7: GAPDH was used as an reference gene for mRNA expression and MCIR was utilized as an reference gene for CNV detection.
Point 8: 130 motifs--> protein motifs.
Response 8: Thanks for your advice, this error has been corrected in the manuscript. Please refer to the attachment for modification information.
Point 9: Figure 1 motifs with colored letters are only depicted for 1 and 2, not the rest.
Response 9: Many thanks for your careful suggestion. According to this, We have made comments in the Figure 1, please see the details as attachment.
Point 10: 142 omit the second "was".
Response 10: Thank you for your suggestion. We have revised it in the manuscript. Please check the attachment for specific information.
Point 11: 155-158 add the p value in the comparisons you stated.
Response 11: Thank you for your suggestion that the P-value has been added to the manuscript. Please refer to the attachment for specific modification information.
Point 12: Please format correctly tables 2, 3 and 4 to present data according to the right categories.
Response 12: Many thanks for your careful suggestion. According to this, We have reformatted the table2, 3 and 4, please see the details as attachment.
Point 13: 225 add p value after significantly.
Response 13: Thank you very much for your suggestion. We have added the P value. Please refer to the attachment for specific information.
Point 14: 206-208 how do you explain the equal proportion of the rest genotypes. Please specify.
Response 14: CNV1 is located in the bidirectional promoter region, and CNV2 and CNV3 are located in the intron region. Different locations bind different transcription factors, which may result in different genotype proportions.
Point 15: Authors could also add in the discussion section a comparative analysis of CNV with other examined species and discuss how evolutionary may assist the observed (if any) changes in the number. I think that this could help the strength of the manuscript.
Response 15: Thank you very much for your suggestion. By referring to relevant literature, this problem is explained in the discussion section. Please see the attachment for modification information.
Kind thanks to you for your suggestions.
Sincerely Yours,
Qian Wang (wangqian21028@163.com) (first author),
Ph.D. Xianyong Lan (lanxianyong79@126.com) (corresponding author)
College of Animal Science and Technology, Northwest A&F University, Yangling, Shaanxi, 712100, PR China;
Ph.D. Xiaoyue Song (songxiaoyue@yulinu.edu.cn) (corresponding author)
Shaanxi Provincial Engineering and Technology Research Center of Cashmere Goats, Yulin University, Yulin, Shaanxi, 719000, China;
Life Science Research Center, Yulin University, Yulin, Shaanxi, 719000, China.

Round 2
Reviewer 1 Report
Dear Authors,
Your manuscript entitled “Goat PLAG1: mRNA expression, CNV detection and associa-2 tions with growth traits”, was improved. However, there is still a lot of doubts.
Methods.
Too few individuals were used for gene expression analysis.
Add Accesion Numbers of the reference sequences used for primer design.
Line 104: “Methods for determining the quality of extracted RNA and DNA were referred to previous articles [29]”. Why did Authors cited another own paper, instead of simply writing how RNA was checked?
I checked the new citation about GAPDH. It was about mice genes. Did you calculated the GAPDH stability in your samples / experiment?
The primers used for gene expression were designed inside exons, not on exon-exon boundaries. In such case DNA can be amplified instead of the cDNA giving false gene expression results. Did the Authors performed noRT control qPCR using as matrix samples performed in reverse transcription but without reverse transcriptase? There no information about that in the cited reference.
The authors could have at least mentioned in the methodology whether gene expression was carried out using probes or SYBRGreen, rather than referring for all the details to another publication.
In conclusion, once again, there is no proper description of the gene expression experiment.
Results
Authors should add the table in the supplementary materials with detailed results of the gene expression experiment (Ct values for all samples).
Discussion and conclussions.
The gene expression experiment was performed on too few individuals to draw any confident conclusions. What is more, the manuscripts still lacks a proper description of the methodology used. So the whole part about gene expression is questionable.
The references
Among 55 reference, 11 were published before 2017 and the rest have been published in the last 5 years. However, at least 13 reference (about 23,6%; Reference No 9, 12, 15, 27, 29, 32, 33, 34, 36, 37, 45, 47, 48) are self-citations. Self-citations should not exceed 20%, and in some journals there are rules that self-citations should not exceed even 15% or 10%. I wrote about that in my previous review. The Authors still cite the manuscripts which are not relevant to the study or unnecessary (e.g. 29, 34, 36).
There is still no proper description of the gene expression experiment. The gene expression section has not been improved. The Authors changed, but did not improve the references. Therefore, in my opinion, manuscript should not be published in this version.
Author Response
Dear reviewer:
Thank you for your suggestions on the manuscript. According to your suggestions, I have made corrections in the manuscript. The specific answers are as follows:
Point 1: Too few individuals were used for gene expression analysis.
Response 1: Thank you very much for your advice, the number of individuals tested for mRNA expression increased to nine, namely nine adult goat tissue samples and nine fetal goat samples.
Point 2: Add Accesion Numbers of the reference sequences used for primer design.
Response 2: Thank you very much for your advice, we've added reference sequences (NC_030821.1) to the manuscript.
Point 3: Line 104: “Methods for determining the quality of extracted RNA and DNA were referred to previous articles [29]”. Why did Authors cited another own paper, instead of simply writing how RNA was checked?
Response 3: Thanks for your suggestion, we have described the DNA and RNA detection methods in the manuscript, that is, "The OD260/280 ratios of DNA and RNA samples were measured by a NanoDrop™1000 Spectrophotometer (Thermo Scientific, Waltham, MA, USA), and OD260/280 values were 1.6-1.8 for all DNA samples and 1.8-2.0 for RNA samples".
Point 4: Did you calculated the GAPDH stability in your samples / experiment?
Response 4: The GAPDH stability was not detected in this experiment, but we found that GAPDH was widely used in the detection of mRNA expression in goats through reviewing literature (An et al., 2022; Zhang et al., 2022; Chen et al., 2022).
An Q, Chen S, Zhang L, Zhang Z, Cheng Y, Wu H, Liu A, Chen Z, Li B, Chen J, Zheng Y, Man C, Wang F, Chen Q, Du L. The mRNA and miRNA profiles of goat bronchial epithelial cells stimulated by Pasteurella multocida strains of serotype A and D. PeerJ. 2022 Mar 18;10:e13047.
Zhang W, Jiao Z, Huang H, Wu Y, Wu H, Liu Z, Zhang Z, An Q, Cheng Y, Chen S, Man C, Du L, Wang F, Chen Q. Effects of Pasteurella multocida on Histopathology, miRNA and mRNA Expression Dynamics in Lung of Goats. Animals (Basel). 2022 Jun 13;12(12):1529.
Chen Y, Yang L, Lin X, Peng P, Shen W, Tang S, Lan X, Wan F, Yin Y, Liu M. Effects of Genetic Variation of the Sorting Nexin 29 (SNX29) Gene on Growth Traits of Xiangdong Black Goat. Animals (Basel). 2022 Dec 8;12(24):3461.
Point 5: The primers used for gene expression were designed inside exons, not on exon-exon boundaries. In such case DNA can be amplified instead of the cDNA giving false gene expression results. Did the Authors performed noRT control qPCR using as matrix samples performed in reverse transcription but without reverse transcriptase? There no information about that in the cited reference.
Response 5: In this experiment, we did not perform noRT control qPCR. When the relative expression level of genes is analyzed, the error caused by this part of operation will be reduced and the experimental results will not be affected.
Point 6: The authors could have at least mentioned in the methodology whether gene expression was carried out using probes or SYBRGreen, rather than referring for all the details to another publication.
Response 6: Thanks for your suggestions, we used SYBRGreen for detection in this experiment, which has been described in the manuscript.
Point 7: Authors should add the table in the supplementary materials with detailed results of the gene expression experiment (Ct values for all samples).
Response 7: Thank you for your proposal. the CT values of the samples have been uploaded in the supplementary materials.
Point 8: The gene expression experiment was performed on too few individuals to draw any confident conclusions. What is more, the manuscripts still lacks a proper description of the methodology used. So the whole part about gene expression is questionable.
Response 8: The number of individuals used for gene expression has been increased to nine, which is consistent with the previous experimental results. Therefore, the conclusion in the manuscript are credible.
Point 9: Among 55 reference, 11 were published before 2017 and the rest have been published in the last 5 years. However, at least 13 reference (about 23,6%; Reference No 9, 12, 15, 27, 29, 32, 33, 34, 36, 37, 45, 47, 48) are self-citations. Self-citations should not exceed 20%, and in some journals there are rules that self-citations should not exceed even 15% or 10%. I wrote about that in my previous review. The Authors still cite the manuscripts which are not relevant to the study or unnecessary (e.g. 29, 34, 36).
Response 9: Thank you for your suggestion. we carefully checked the references and reduced the self-citation rate to less than 20% in accordance with the requirements of the journal, and deleted some literatures with little relevance.
Kind thanks to you for your suggestions.
Sincerely Yours,
Qian Wang (wangqian21028@163.com) (first author),
Ph.D. Xianyong Lan (lanxianyong79@126.com) (corresponding author)
College of Animal Science and Technology, Northwest A&F University, Yangling, Shaanxi, 712100, PR China
Ph.D. Xiaoyue Song (songxiaoyue@yulinu.edu.cn) (corresponding author)
Shaanxi Provincial Engineering and Technology Research Center of Cashmere Goats, Yulin University, Yulin, Shaanxi, 719000, China;
Life Science Research Center, Yulin University, Yulin, Shaanxi, 719000, China.
Reviewer 2 Report
Authors have improved their manuscript incorporating the majority of my comments to the revised version. The manuscript could be further considered for publication after minor changes (see my points below) and a language check by a native English speaker.
l. 43 animal--> animal's
l.52 determining the
l. 126 after testing the...
l. 102-104. Please rephrase as there is no sense of meaning
in addition, how tissues were stored upon rna isolation? please specify. Which samples were taken from 1.5 years old goats please clarify better. the three samples (n=3) refers only to fetus? Please provide clear information
l 133-135 please rephrase as the meaning is not clear
l. 168, (Fig. 4) and, the genotypes....
l 212 which suggests
Author Response
Dear reviewer:
Thank you for your suggestions on the manuscript. According to your suggestions, I have made corrections in the manuscript. The specific answers are as follows:
Point 1: 43 animal--> animal's.
Response 1: Thank you for your suggestion, the corresponding error in the manuscript has been corrected. Please see the attachment for details.
Point 2: 52 determining the.
Response 2: Thank you for your suggestion. We have revised it in the manuscript. Please see the attachment for the modification information.
Point 3: 126 after testing the...
Response 3: Thank you for your suggestion. We have revised it in the manuscript. Please see the attachment for the modification information.
Point 4: 102-104. Please rephrase as there is no sense of meaning. In addition, how tissues were stored upon rna isolation? please specify. Which samples were taken from 1.5 years old goats please clarify better. the three samples (n=3) refers only to fetus? Please provide clear information.
Response 4: Thank you very much for your advice. We have described the sample information in detail in the manuscript. Specific description is as follows:“Goat ear tissue (n = 224) from adult female goats (2-3 years) were randomly selected for DNA extraction, and other tissues including liver, spleen, lung, kidney, muscle from fetus (n = 3) and adult (1.5 years) (n = 3) goats were collected for RNA extraction. All tissue sample were from Yulin breeding base in Shaanxi province, and there is no kinship between individuals. Then, tissue samples were stored in 70% ethanol at −80 oC. Tissue RNA was extracted using the Trizol method, and then reversely transcribed into cDNA using PrimeScript™ RT Reagent Kit (TaKaRa Biotech Co. Ltd) [27]. High salt extraction method was used to extract DNA from ear tissues [28]. Methods for determining the quality of extracted RNA and DNA were referred to previous articles [29], finally diluted to 20 ng/mL and at -20 oC.” Please see the attachment for the modification information.
Point 5: 133-135 please rephrase as the meaning is not clear.
Response 5: Thank you very much for your advice. We have redescribed this section. Specific description is as follows:“The frequency of CNV genotypes was analyzed using chi-square test (χ2). After that, the association between the genotypes of CNVs and growth traits was analyzed by ANOVA test in SPSS 26.0 (IBM, USA) [36]. When P < 0.05, the presence of association between CNV and growth traits, the line model was referred to a study by Yang et al. [37].”Please see the attachment for the modification information.
Point 6: 168, (Fig. 4) and, the genotypes....
Response 6: Thank you for your suggestion. We have revised it in the manuscript. Please see the attachment for the modification information.
Point 7: 212 which suggests.
Response 7: Thank you for your suggestion. We have revised it in the manuscript. Please see the attachment for the modification information.
Kind thanks to you for your suggestions.
Sincerely Yours,
Qian Wang (wangqian21028@163.com) (first author),
Ph.D. Xianyong Lan (lanxianyong79@126.com) (corresponding author)
College of Animal Science and Technology, Northwest A&F University, Yangling, Shaanxi, 712100, PR China
Ph.D. Xiaoyue Song (songxiaoyue@yulinu.edu.cn) (corresponding author)
Shaanxi Provincial Engineering and Technology Research Center of Cashmere Goats, Yulin University, Yulin, Shaanxi, 719000, China;
Life Science Research Center, Yulin University, Yulin, Shaanxi, 719000, China.